# Direct catalytic asymmetric synthesis of α-chiral bicyclo[1.1.1]pentanes

Marie L. J. Wong[1], Alistair J. Sterling [1], James J. Mousseau[2], Fernanda Duarte [1✉] & Edward A. Anderson [1✉]

Bicyclo[1.1.1]pentanes (BCPs) are important motifs in contemporary drug design as linear spacer units that improve pharmacokinetic profiles. The synthesis of BCPs featuring adjacent stereocenters is highly challenging, but desirable due to the fundamental importance of 3D chemical space in medicinal chemistry. Current methods to access these high-value chiral molecules typically involve transformations of pre-formed BCPs, and can display limitations in substrate scope. Here we describe an approach to synthesize α-chiral BCPs involving the direct, asymmetric addition of simple aldehydes to [1.1.1]propellane, the predominant BCP precursor. This is achieved by combining a photocatalyst and an organocatalyst to generate a chiral α-iminyl radical cation intermediate, which installs a stereocenter simultaneously with ring-opening of [1.1.1]propellane. The reaction proceeds under mild conditions, displays broad scope, and provides an array of α-chiral BCPs in high yield and enantioselectivity. We also present a theoretical model for stereoinduction in this mode of photoredox organocatalysis.

[1] Chemistry Research Laboratory, 12 Mansfield Road, Oxford OX1 3TA, UK. [2] Pfizer Medicine Design, Eastern Point Road, Groton CT 06340, USA. ✉email: fernanda.duartegonzalez@chem.ox.ac.uk; edward.anderson@chem.ox.ac.uk

Bicyclo[1.1.1]pentanes (BCPs) have become established as useful bioisosteres for 1,4-disubstituted arenes[1–5], alkynes[6], and *tert*-butyl[7] groups in the design of functional molecular scaffolds, particularly in drug discovery (Fig. 1a)[8,9]. Compounds containing this rigid three-dimensional motif, which often exhibit improved pharmacochemical profiles compared to their parent functionalities[10], are most commonly accessed by ring-opening reactions of the inter-bridgehead bond of [1.1.1]propellane (**1**, tricyclo[1.1.1.0$^{1,3}$]pentane)[11–14]. Despite many recent advances in the preparation of BCPs[15–28], the asymmetric synthesis of α-chiral variants (which would serve as surrogates for benzylic, propargylic, or neopentylic stereocenters) is highly challenging; current approaches entail lengthy synthetic routes involving chemical resolution processes[29] or the use of stoichiometric chiral auxiliaries/reagents (Fig. 1b)[17,18,30,31]. Only two methods have been described that enable the catalytic asymmetric synthesis of α-chiral BCPs (Fig. 1c); these consist of a rhodium-catalyzed C–H insertion of α-diazoesters into the bridgehead C–H bond of 3-aryl-BCPs to give α-chiral BCP esters[32], and an iridium-catalyzed asymmetric allylation of BCP-zinc complexes[33] (formed in situ from addition of aryl Grignards to **1** at elevated temperatures)[6] to form enantioenriched allyl-BCP derivatives.

The enantioselective synthesis of α-chiral BCPs via stereoselective ring-opening reactions with **1** represents an attractive route to access these high-value compounds. As the addition of radicals to **1** is typically highly efficient and proceeds under mild conditions[21,23], we questioned whether a direct asymmetric radical addition could simultaneously generate the BCP motif and the adjacent stereocenter (Fig. 1d). Enantioselective α-carbonyl functionalizations of aldehydes through singly occupied molecular orbital (SOMO) activation have been shown to be highly selective and efficient radical reactions for a range of asymmetric transformations[34,35]. Furthermore, the combination of this strategy with photoredox catalysis enables these reactions to proceed with simple, unactivated coupling partners, and without the need for stoichiometric oxidants[36–39]. We envisaged that this type of multicatalytic strategy, employing a photocatalytically generated α-iminyl radical cation as a key intermediate[36,40–43], could be utilized to access a wide variety of α-chiral BCPs directly from [1.1.1]propellane.

Our design for this catalytic route to α-chiral BCPs first involves condensation of an aldehyde with a chiral organocatalyst to form an electron-rich enamine **2** (Fig. 2), which then undergoes single-electron oxidation by an excited-state photocatalyst to generate an α-iminyl radical cation **3** (refs. [36,40–43]). This radical cation could

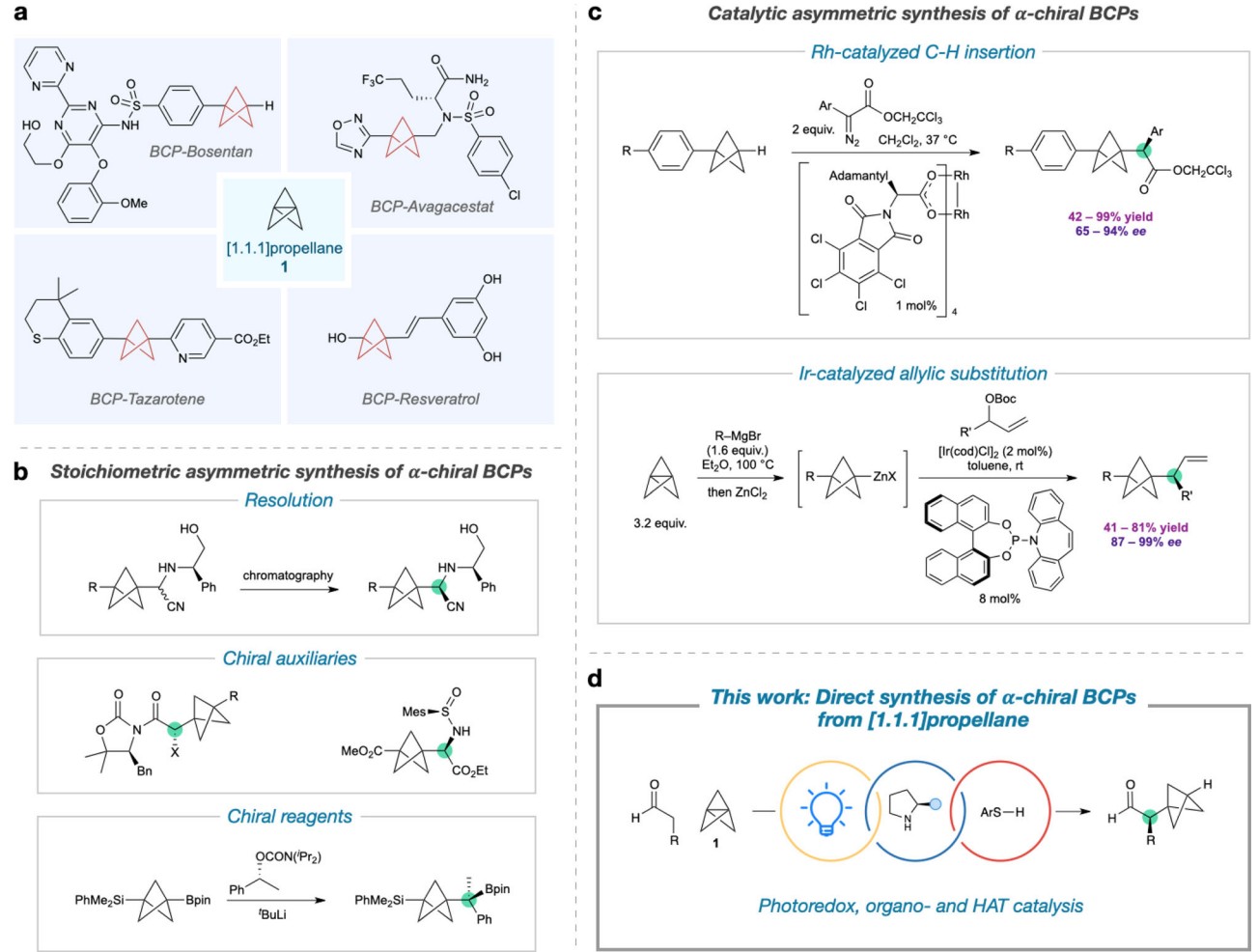

**Fig. 1 Synthesis of α-chiral bicyclo[1.1.1]pentanes. a** Examples of bicyclo[1.1.1]pentanes (BCPs) in pharmaceutical research. The near-ubiquitous precursor is [1.1.1]propellane (**1**). **b** Current approaches to α-chiral bicyclo[1.1.1]pentanes typically require the use of chemical resolution, or stoichiometric chiral auxiliaries/reagents. **c** Catalytic asymmetric synthesis of α-chiral BCPs is possible through insertion of α-diazoesters into BCP bridgehead C–H bonds, or iridium-catalyzed allylic substitution by BCP-zinc complexes formed from ring-opening of **1** with Grignard reagents under thermal conditions. **d** This work: design of a multicatalytic asymmetric radical addition to [1.1.1]propellane to directly access α-chiral BCPs, where the stereocenter and BCP are formed simultaneously on ring opening of **1**.

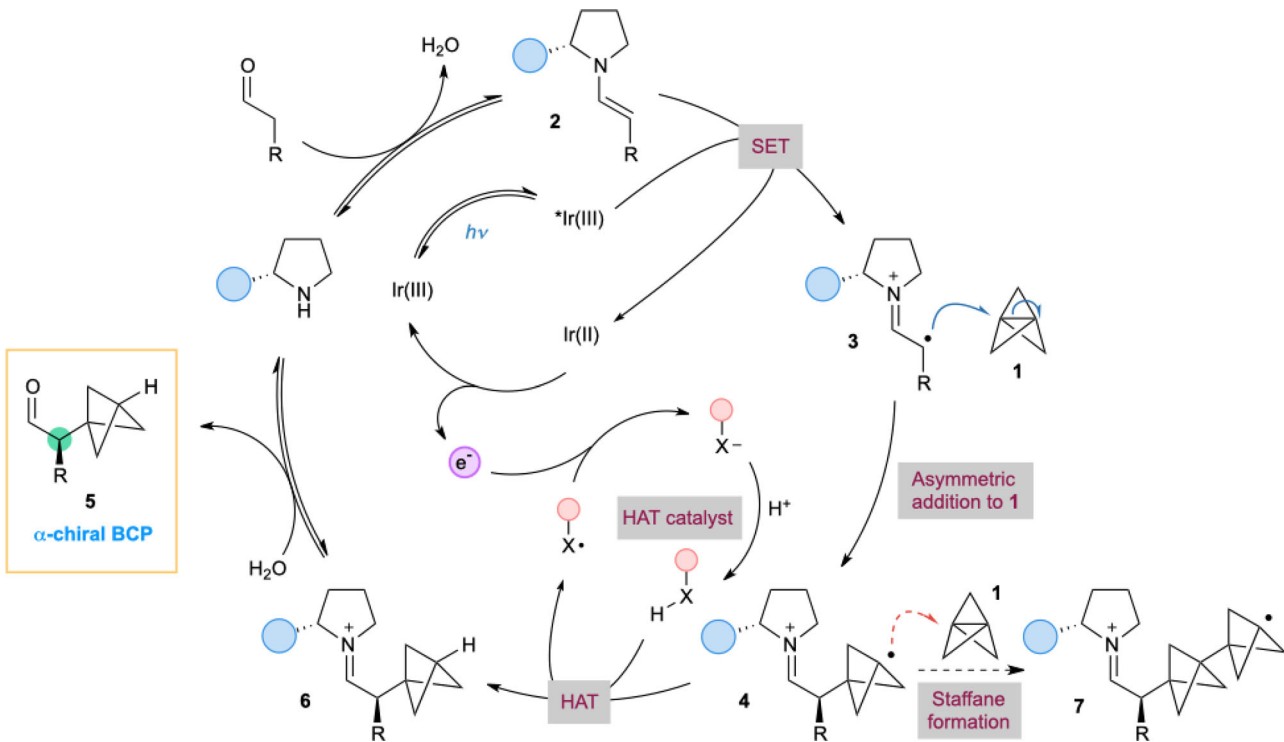

**Fig. 2 Design of a multicatalytic direct asymmetric radical addition to [1.1.1]propellane (1).** Potential side reactions include staffane formation by addition of radical **4** to **1**, and reaction of the HAT catalyst with **1**.

undergo addition to the central C–C bond of **1** to construct the BCP and set the adjacent stereocenter in the same step (**4**). This process would depend critically on the SOMOphilic character of **1**, as aside from heterodimerization of radicals generated from electron donor–acceptor complexes[42,43], only activated alkenes (styrenes)[36] or stable radicals (TEMPO)[40,41] have been shown to react efficiently with chiral α-iminyl radical cations. To complete the organo- and photoredox catalytic cycles, a third process would be incorporated whereby the tertiary BCP radical **4** undergoes a reductive hydrogen atom transfer (HAT)[37,44,45], affording the enantioenriched product **5** after hydrolysis of iminium ion **6**. The radical derived from the HAT catalyst can then oxidize the photocatalyst, regenerating the ground state complex; the HAT catalyst would be reformed via protonation by a proton source such as the organocatalyst pyrrolidinium ion. It is crucial that these three catalytic cycles operate simultaneously and in a highly selective manner, as a number of undesirable side reactions could take place; these include reactions between the different catalysts (in particular, the HAT catalyst) and [1.1.1]propellane itself, and interception of the bicyclopentyl radical **4** by further equivalents of [1.1.1]propellane, leading to oligomeric bicyclo[1.1.1]pentanes (staffanes, **7**)[14]. Here we describe the successful development of this methodology, which offers an efficient method for the direct asymmetric synthesis of α-chiral BCPs from ring-opening of [1.1.1]propellane. The reaction exhibits broad scope and affords excellent levels of enantioselectivity, enabling the synthesis of a range of previously inaccessible enantioenriched BCP products that are of high value for pharmaceutical research.

## Results

**Reaction optimization.** Our investigations began with the reaction of aldehyde **8** and two equivalents of [1.1.1]propellane (as a solution in Et$_2$O) in the presence of the photocatalyst Ir[(ppy)$_2$(dtbbpy)]PF$_6$ (2 mol%), organocatalyst **9** (25 mol%), and HAT catalyst **10** (10 mol%), with 1,2-dimethoxyethane (DME) as

a co-solvent (2:1 DME:Et$_2$O, 0.2 M). Under blue LED irradiation (LED strips), we were pleased to find that the desired BCP product **11** was obtained with high enantioselectivity (89% *ee*), albeit in low yield (Fig. 3a Table, Entry 1); reduction of the aldehyde product to the corresponding alcohol was carried out to facilitate purification and avoid potential epimerization of the newly formed stereocenter. Use of a more powerful 18 W blue LED lamp, and a lower reaction temperature (10 °C) dramatically improved the yield (Entry 2). An extensive survey of reaction variables was then carried out (entries 3–5 and panels b–d; see Supplementary Tables 1–7, for full details). We first found that thiols outperformed other classes of HAT catalysts (Fig. 3b, reactions run at 50 mol% loading of HAT catalyst), including several that have been widely used as H-atom sources in other radical reactions. Sterically hindered thiophenol **10** remained the optimal HAT catalyst from this screen; notably the thiol-BCP adduct resulting from the direct reaction of **10** with [1.1.1] propellane[24] was not observed under the reaction conditions (see Supplementary Fig. 1). We next evaluated a range of organocatalysts (Fig. 3c, reactions run at 25 mol% loading of organocatalyst), which revealed that the size of the silyl ether substituent (OTMS), and the nature of the arene substitution (3,5-CF$_3$) of the diarylprolinol were crucial for high yield and selectivity. The choice of photocatalyst (Fig. 3d) was also of key importance, with iridium complexes featuring a dtbbpy (4,4′-di-*tert*-butyl-2,2′-bipyridine) ligand proving most effective. The reaction performed well in ethereal solvents (Fig. 3d, 2:1 ratio of co-solvent to Et$_2$O from [1.1.1]propellane solution), and a mixture of DME and Et$_2$O (2:1 or 1:1) was identified as ideal to ensure solubility of the reactants (see Supplementary Table 5 for further discussion). Employing **1** as the limiting reactant led to an additional improvement in yield, while maintaining high enantioselectivity (Entries 6 and 7). A series of control experiments confirmed that all three catalysts were required, with no or only trace reaction observed with the omission of any catalyst (Entries 8–10). Under the optimized reaction conditions (Entry 7), we obtained the

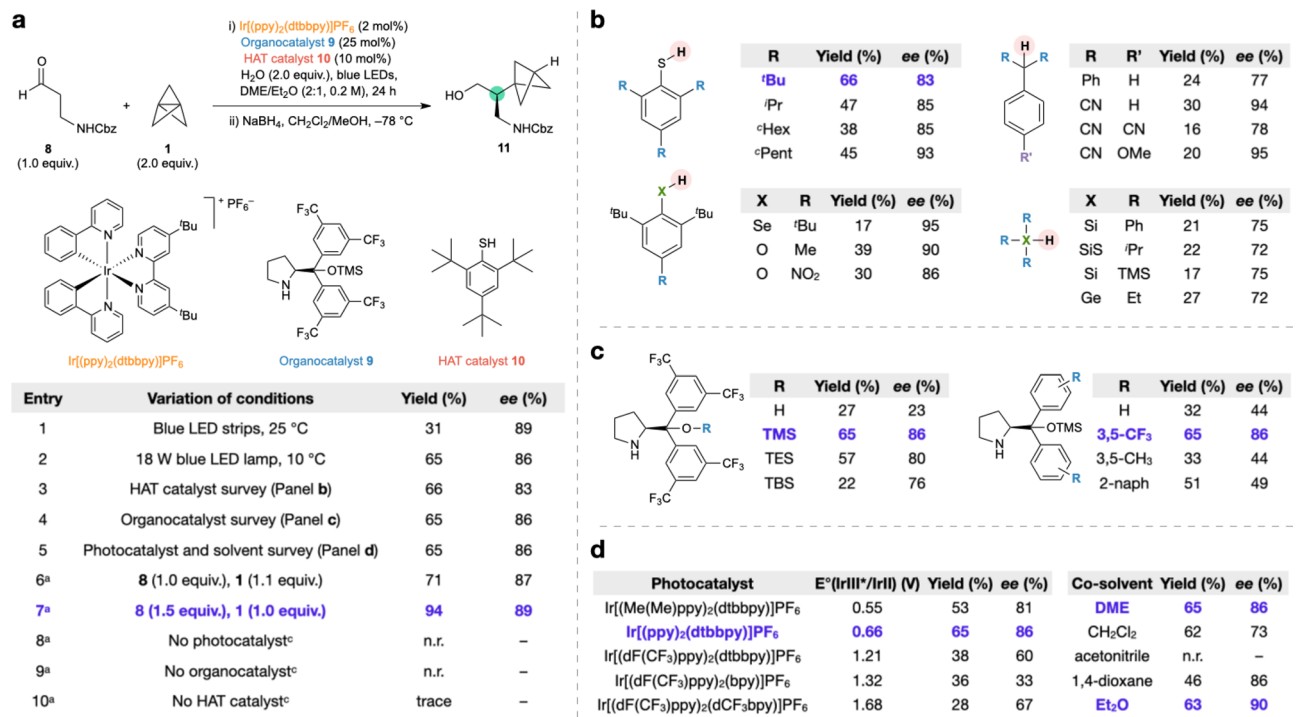

**Fig. 3 Optimization of the asymmetric addition to [1.1.1]propellane. a** Optimization of reaction conditions. Entries 2-10 were conducted using an 18 W blue LED lamp at 10 °C. All yields are isolated yields. [a]DME:Et$_2$O (1:1, 0.2 M). **b** A survey of H-atom sources (50 mol% loading) shows HAT catalyst **10** is optimal. **c** A survey of organocatalysts (25 mol% loading) shows catalyst **9** is optimal. **d** A survey of photocatalysts shows a range of iridium photocatalysts are effective; see the Supplementary Table 4 for details of reduction potentials. Co-solvents were screened as a 2:1 ratio of co-solvent to Et$_2$O (from the solution of **1**), with an overall concentration of 0.2 M. For full details of reaction optimization, see Supplementary Tables 1-7. n.r. no reaction.

α-chiral BCP product **11** in 89% *ee* and 94% yield based on [1.1.1] propellane. As **1** is the most valuable component of the reaction system, and has the potential to undergo a number of side reactions, this efficiency is notable.

**Substrate scope**. We found that this asymmetric construction of α-chiral BCPs was successful with aldehyde substrates bearing a wide range of functional groups (Fig. 4a). The α-functionalized products were generally obtained in high yields, and with excellent levels of enantioselectivity. Simple alkyl aldehydes performed well in the reaction (**12–15**, 63–96%, 64–98% *ee*), including more sterically demanding substrates **13** and **14**, although only modest selectivity was obtained when the aldehyde α-carbonyl substituent was a methyl group (**12**, 63%, 64% *ee*). Aldehydes bearing aryl (**16–18**, 72–99%, 90–96% *ee*), alkenyl (**19**, 70%, 90% *ee*), and alkynyl (**20**, 50%, 90% *ee*) moieties also afforded the corresponding BCP products in good yields and high selectivities. Substrates featuring various heteroatom functionalities, including thioethers, carbamates, halides, esters, and ethers were successful in the reaction, again delivering BCP products with high efficiency and enantioselectivity (**11**, **21–26**, 83–99%, 86–97% *ee*). Particularly notable is the formation of enantioenriched BCP piperidine **25**, a class of heterocyclic product that is of importance in pharmaceutical research. Substrates bearing heteroaromatic motifs were found to be more challenging: while furanyl-BCP **27** was obtained in 89% yield and 95% *ee*, the formation of thiophenyl and pyridinyl products **28** and **29** was lower yielding, albeit proceeding with excellent enantioselectivities (25–29%, 90–96% *ee*). The reaction can be successfully scaled up with no reduction in yield or enantioselectivity, with alcohol **16** prepared in 98% yield and 96% *ee* on a 1 mmol scale. To further expand the potential of this chemistry, we considered other methods to trap the bicyclopentyl radical resulting from addition of the α-iminyl radical cation to **1**. Using *N*-(phenylthio)phthalimide instead of the HAT catalyst, we were able to obtain sulfur-substituted BCP product **30** in 50% yield and 88% *ee*; this reaction presumably proceeds by attack of the BCP radical on the S–N bond, leading to a phthalimidyl radical that is also capable of catalyst turnover ($E°$ [Ir(III)/Ir(II)] = –1.51 V vs. SCE[46]; $E°$[SuccN•/SuccN⁻] of the related succinimidyl radical has been measured as +1.96 V vs. SCE)[47]. These results show that this methodology offers a versatile route to access α-chiral bicyclo[1.1.1]pentanes containing a wide variety of functional groups directly from [1.1.1]propellane and commercially available or easily accessible starting materials and catalysts.

**Product derivatization**. To demonstrate the utility of the α-chiral BCP products, we explored transformations of the enantioenriched aldehyde obtained directly from the photoredox reaction (Fig. 4b). Aldehyde **31** could be oxidized to the carboxylic acid without racemization of the stereocenter (**32**, 97%, 95% *ee*), and also underwent reductive amination with benzylamine to afford secondary amine **33** in 93% yield and 92% *ee*. The reaction of **31** with an aryl Grignard reagent proceeded with excellent diastereoselectivity, giving secondary alcohol **34** in 95% yield (94% *ee*, 14:1 *dr*) under Felkin-Anh control. Lastly, **31** could be homologated to alkyne **35**, again without erosion of enantiopurity (97%, 96% *ee*). Such α-chiral BCP products are likely to be of high value for various applications.

**Mechanistic studies and DFT calculations**. As discussed above, we propose that the mechanism of this reaction involves initial oxidation of the enamine to the α-iminyl radical cation by the excited state Ir(III) complex ($E°$[Ir(III)*/Ir(II)] = +0.66 V vs. SCE[46]; $E°$[α-iminyl radical cation/propanal enamine] = +0.77 V vs. SCE[48]; $\Delta E° = –0.11$ V). Stern-Volmer fluorescence

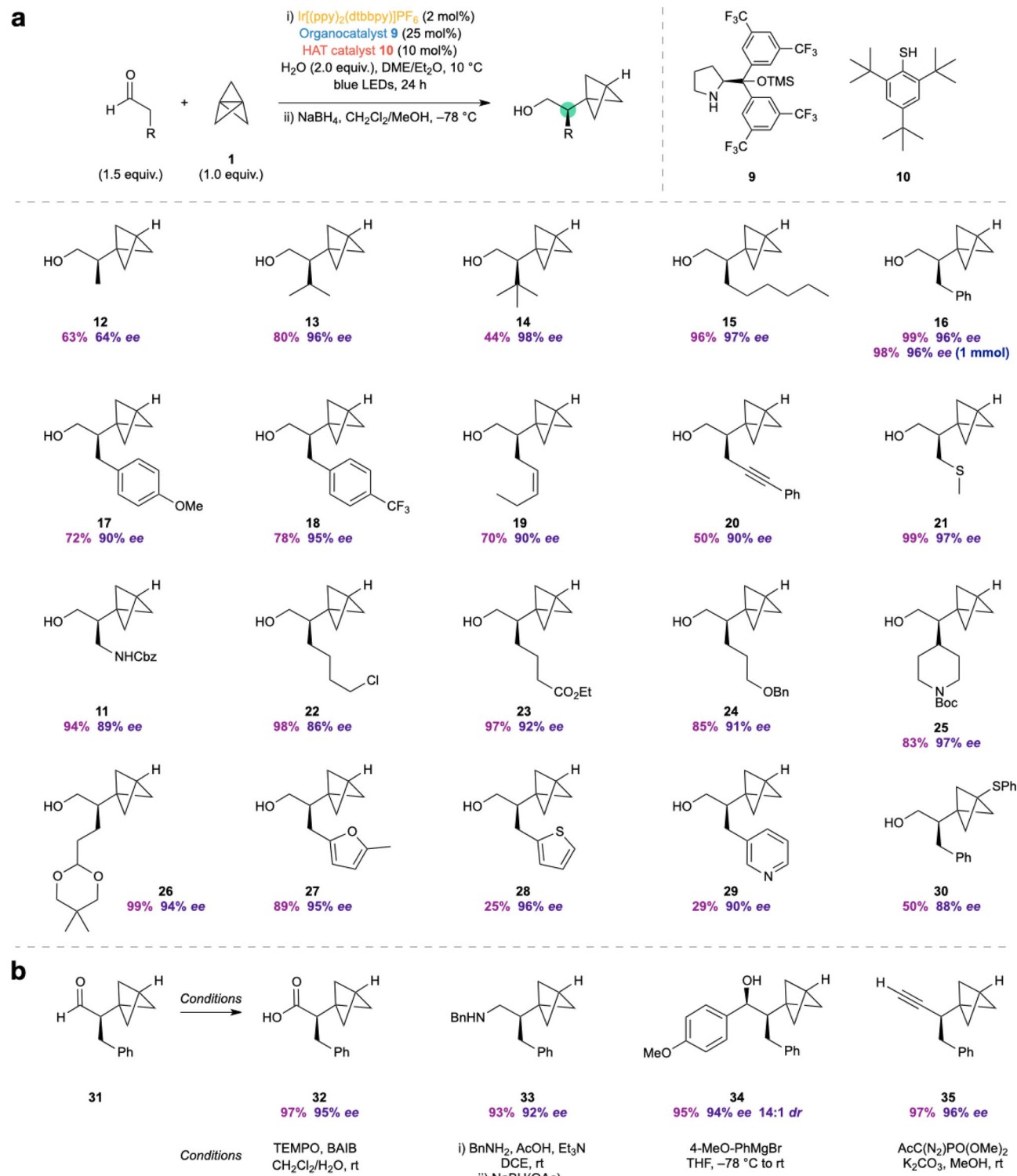

**Fig. 4 Scope of the reaction and product derivatization. a** The catalytic asymmetric synthesis of α-chiral BCPs displays a wide substrate scope and functional group tolerance, affording enantioenriched BCP products in high yields and enantioselectivities. **b** The aldehyde product can be derivatized to a range of other high-value chiral BCP building blocks. The absolute stereochemistry of **12** was assigned by comparison of supercritical fluid chromatography (SFC) data of a derivative with its enantiomer[30]; other products were assigned by analogy. See Supplementary Information, Section 3.2.2 for details. TEMPO = (2,2,6,6-tetramethylpiperidin-1-yl)oxyl; BAIB = PhI(OAc)$_2$; DCE = 1,2-dichloroethane.

quenching experiments reveal the enamine to be the most efficient quencher of this photoexcited Ir(III) complex, with a quenching constant of 83.4 M$^{-1}$ – 10 times more efficient than the next-best quencher in the reaction (see Supplementary Information, Section 5). Addition of the α-iminyl radical cation

to **1** leads to a bicyclopentyl radical, which abstracts a hydrogen atom from the thiol HAT catalyst to deliver the BCP product; the thiyl radical that results is a competent species for catalyst turnover ($E°$[PhS•/PhS$^-$] = +0.16 V vs. SCE[49]; $E°$[Ir(III)/Ir(II)] = −1.51 V vs. SCE[46]; Δ$E°$ = +1.67 V). Finally, the sulfide ion is

## a  General kinetic model for the stereodetermining step of the reaction

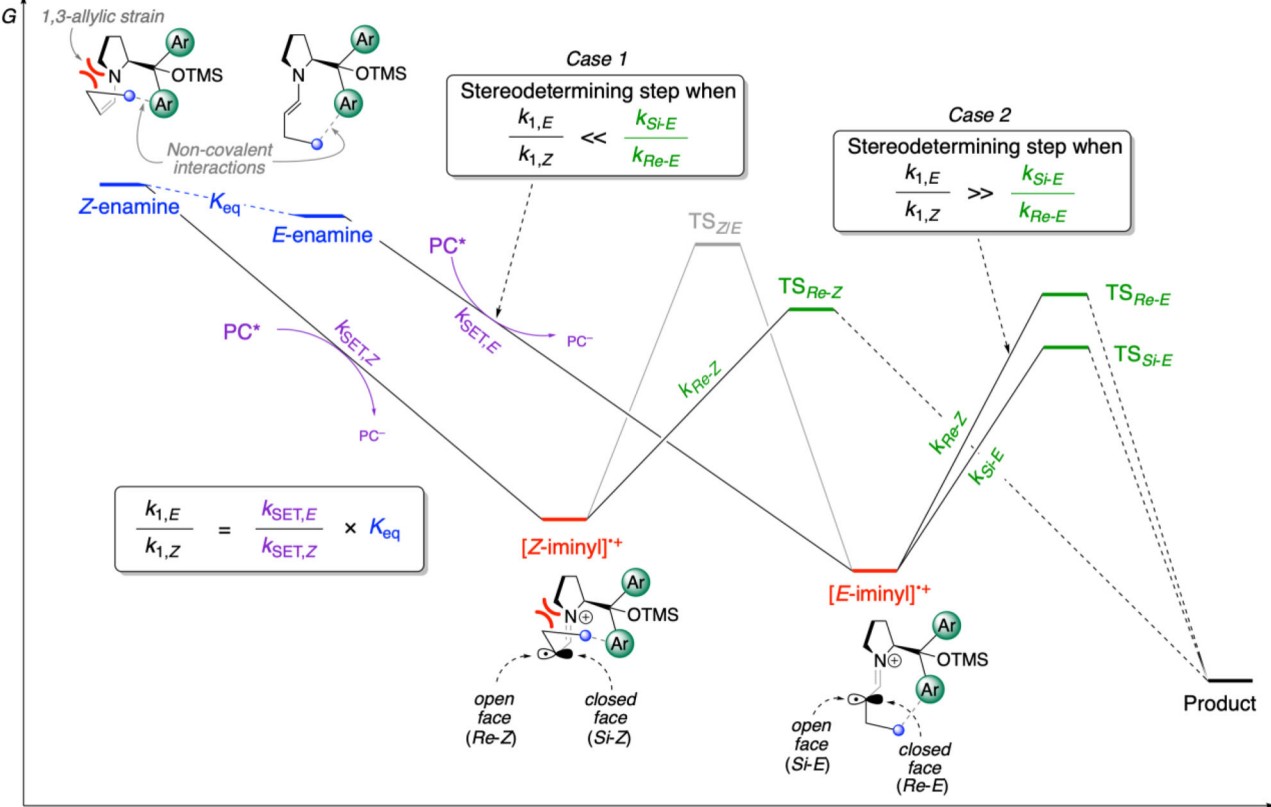

## b  Key substrate–diarylprolinol non-covalent interactions

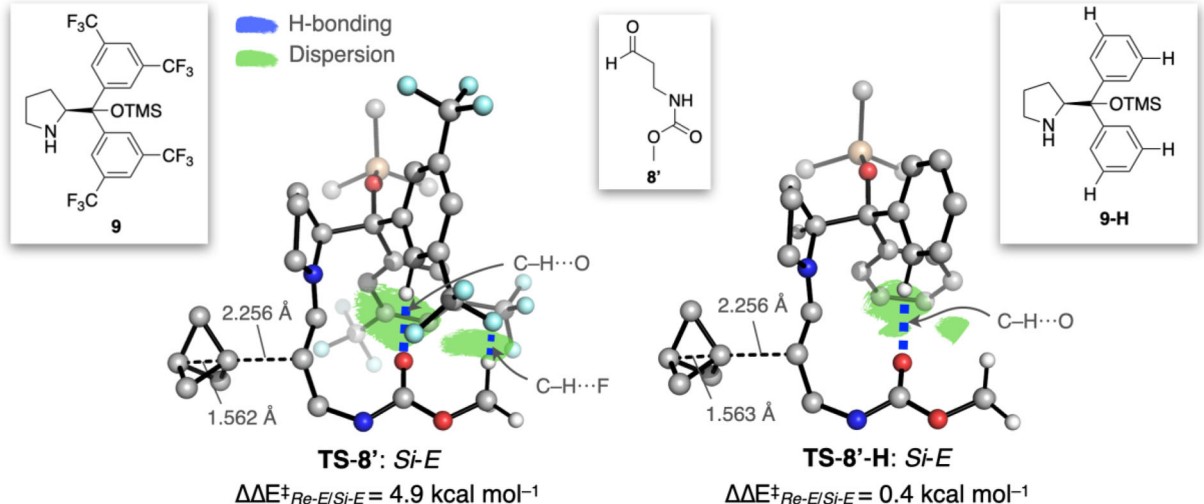

**TS-8': Si-E**
ΔΔE‡$_{Re-E/Si-E}$ = 4.9 kcal mol⁻¹

**TS-8'-H: Si-E**
ΔΔE‡$_{Re-E/Si-E}$ = 0.4 kcal mol⁻¹

**Fig. 5 Energy profiles for key species on the proposed reaction pathway and transition state geometries for addition of the iminyl radical cation to 1.**
**a** Populations of, and transitions between, key species involved in the stereodetermining steps of the reaction. PC photocatalyst, SET single-electron transfer. **b** Lowest energy constrained transition states for *Si-E* **TS-8'** and **TS-8'-H**, showing the most important non-covalent interactions between the sidechain and pyrrolidine substituents, calculated at the [SMD(DME)-B2GP-PLYP-D3BJ/def2-TZVPP//SMD(DME)-PBE0-D3BJ/def2-SVP] level of theory. Hydrogen atoms are omitted for clarity. Blue = hydrogen bonding. Green = dispersion interactions.

reprotonated by an acid such as the pyrrolidinium ion derived from organocatalyst **9**.

While this sequence offers a viable pathway for product formation, our optimization work showed that the enantioselectivity of the reaction is dependent on a number of factors, including the nature of the aldehyde sidechain, the pyrrolidine organocatalyst substituents, and the photocatalyst. To rationalize this, we turned to computational analysis (for a full discussion of the methodology and models employed, see the Supplementary Information, Section 4). While enamine isomerization via the iminium ion is likely to be rapid, calculations on a model system revealed *E/Z* isomerization of the α-iminyl radical cation itself to be slow, with a barrier higher than the subsequent trapping of **1** (see Supplementary Fig. 3). Addition of **1** to either the *E*- or *Z*-α-iminyl radical cation can

potentially occur from either the 'open' or 'closed' face (Fig. 5a), resulting in four pathways (*Si-E*, *Re-E*, *Re-Z*, and *Si-Z*) that lead to two diastereomers of the BCP radical product. However, stabilizing non-covalent interactions (NCIs) between the substrate sidechain and the electron-deficient diarylprolinol substituents enhance shielding of the closed face of the α-iminyl radical cation. While formation of the major enantiomer could thus occur through either a *Si-E* or *Si-Z* transition state (TS), the latter pathway is strongly disfavored due to both 1,3-allylic strain, and the abovementioned steric hindrance to closed-face attack, and therefore has a negligible contribution.

Due to the slow isomerization between the *E*- and *Z*-iminyl radical cations, the observed stereoselectivity could be influenced by the relative rates of (irreversible) oxidation of the two enamines, and/or by the facial selectivity of addition of **1** to the *E*-α-iminyl radical cation. In the case where addition to the closed face is highly disfavored due to substrate/diarylprolinol NCIs (such that reaction occurs only via the open face *Si-E* or *Re-Z* pathways), the enantiomeric ratio (*er*) is then equal to the ratio of observed enamine oxidation rates ($k_{1,E}/k_{1,Z}$, Case 1, Fig. 5a). This is defined by the ratio of intrinsic electron transfer rates, $k_{SET,E}/k_{SET,Z}$ multiplied by the equilibrium constant for the *E/Z* enamine population ($K_{eq}$ = [*E*-enamine]/[*Z*-enamine]); in other words, $(k_{1,E}/k_{1,Z}) \ll (k_{Si-E}/k_{Re-E})$. However, where weaker NCIs are present, the energy difference between open and closed face addition is less such that $(k_{1,E}/k_{1,Z}) \gg (k_{Si-E}/k_{Re-E})$, and the observed *er* is approximated by the ratio of *Si-E* and *Re-E* rate constants ($k_{Si-E}/k_{Re-E}$, Case 2). If $(k_{1,E}/k_{1,Z})$ and $(k_{Si-E}/k_{Re-E})$ are similar in magnitude, the minor enantiomer will be formed through both the *Re-Z* and *Re-E* pathways.

To investigate these factors, we calculated the equilibrium constant ($K_{eq}$) for the enamines derived from organocatalyst **9** and substrate **8′** – a methyl carbamate (Moc) analog of substrate **8** used for reaction optimization (94%, 89% *ee*) – and *Si-E* and *Re-E* TSs for the addition of **1** to the resulting *E*-iminyl radical cations. To obtain a model for each TS, the forming C–C distance was constrained to that obtained from calculations on a simplified system with a large basis set, followed by exhaustive conformational sampling to obtain the lowest energy transition states for the formation of each diastereomer of the product (for a validation of this approach see Supplementary Fig. 6). $K_{eq}$ was found to favor the *E*-enamine by $\Delta G_{E/Z}$ = 3.9 kcal mol⁻¹ ($K_{eq}$ = 989 at 283 K). The subsequent *Si-E* pathway is favored over the *Re-E* pathway by 4.9 kcal mol⁻¹ ($k_{Si-E}/k_{Re-E}$ = 5612 at 283 K), and as a result the stereoselectivity for this substrate is described by Case 1. The strong preference for open face addition arises from stabilizing C–H···F and C–H···O NCIs between the α-iminyl radical cation sidechain and the diarylprolinol substituents (Fig. 5b). Notably, polar functionality is not a prerequisite for high selectivity: a simple linear alkyl sidechain on the aldehyde also provides sufficient favorable dispersion and C–H···F interactions to avoid closed-face *Re-E* addition (see Supplementary Fig. 12), a finding that is consistent with the excellent *ee* observed for the reaction with octanal (97%, product **15**).

When the trifluoromethyl groups of organocatalyst **9** are replaced with hydrogen atoms (i.e. a phenyl ring, **9-H**), substrate–diarylprolinol NCIs are weakened. The major enantiomer still arises from open face *Si-E* attack (**TS-8′-H**, Fig. 5b), but closed face (*Re-E*) addition is disfavored by only 0.4 kcal mol⁻¹ ($k_{Re-E}/k_{Si-E}$ = 2.2 at 283 K, 34% *ee*$_{calc}$), offering a low-energy pathway for the formation of the minor enantiomer. This predicted value supports the experimentally observed enantioselectivity using this organocatalyst (see Fig. 3c, 44% *ee*). Here the *Si-E/Re-Z* pathway competition, which is controlled by the $k_{1,E}/k_{1,Z}$ ratio, becomes inconsequential ($\Delta G_{E/Z}$ = 4.8 kcal mol⁻¹, $K_{eq}$ = 5092 at 283 K).

For case 1 (substrate **8** or **8′**/organocatalyst **9**), we initially expected the rates of *E*- and *Z*-enamine oxidation to be approximately equal such that only $K_{eq}$ affects the enantioselectivity. However, this scenario leads to a predicted *ee* of 99.8% for the reaction with substrate **8′**. The experimentally observed *ee* of 89% appears to suggest that oxidation of the *Z*-enamine is ~50 times faster than the *E*-enamine ($k_{SET,E}/k_{SET,Z}$ = 0.017) – a phenomenon that could explain the variation in enantioselectivity with the photocatalyst, as the oxidation rates of the *Z*- and *E*-enamines would depend on the exact nature of the photocatalyst used. A more rapid oxidation of the *Z*-enamine may indeed be a reasonable expectation, as upon oxidation its C = C bond lengthens by 0.05 Å, which relieves 1,3-allylic strain. A similar analysis of the origins of the low observed enantioselectivity for the reaction with propanal (64% *ee*) reveals that while NCIs between its methyl sidechain and diarylprolinol are weakened compared with **8′** or octanal (Supplementary Fig. 11), closed face addition is still disfavored ($\Delta\Delta E^{\ddagger}_{Si-E/Re-E}$ = 3.6 kcal mol⁻¹, $k_{Si-E}/k_{Re-E}$ = 631 at 283 K). Selectivity is again predicted to be determined by the *E/Z* enamine equilibrium ($\Delta G_{E/Z}$ = 3.1 kcal mol⁻¹, $K_{eq}$ = 246 at 283 K, case 1) and subsequent oxidation.

Collectively, these results show that both strongly favoring the *E*-enamine geometry and the formation of hydrogen bonds/dispersion interactions between the substrate sidechain and the diarylprolinol substituents are essential to achieve high enantioselectivity. These effects act synergistically, increasing the bias for formation of the *E*-iminyl radical cation and its subsequent attack by [1.1.1]propellane **1** from the open face of the radical cation.

In conclusion, we have developed an efficient and enantioselective synthesis of α-chiral bicyclo[1.1.1]pentanes, exploiting a multicatalytic system that incorporates photoredox, organo-, and HAT catalysis. The reaction tolerates substrates bearing saturated and unsaturated carbon chains, heterocyclic motifs, aryl and heteroaryl groups, and heteroatoms, and offers a general means to synthesize enantioenriched α-chiral BCPs directly from [1.1.1]propellane. The products obtained can then be easily diversified, allowing the synthesis of valuable chiral BCP building blocks that were previously impossible or difficult to obtain, and highlighting the potential for this protocol to be employed in a wide variety of synthetic applications. We have also developed a kinetic model that rationalizes stereochemical outcomes in this reaction class, where non-covalent interactions play a crucial role in achieving high enantioselectivity.

## Methods

**Typical procedure for asymmetric addition of aldehydes to [1.1.1]propellane.** To a flame-dried vial was added Ir[(ppy)₂(dtbbpy)]PF₆ (3.7 mg, 4.0 μmol, 0.02 equiv.), 2,4,6-tri-*tert*-butylbenzenethiol **10** (5.6 mg, 0.02 mmol, 0.1 equiv.), (*S*)-α,α-bis[3,5-bis(trifluoromethyl)phenyl]-2- pyrrolidinemethanol trimethylsilyl ether **9** (30 mg, 0.050 mmol, 0.25 equiv.), and aldehyde (0.30 mmol, 1.5 equiv., if solid). The vial was sealed with a rubber septum and Et₂O, DME, distilled water (7.2 μL, 0.40 mmol, 2.0 equiv.) were then added (taking into account Et₂O from the solution of [1.1.1]propellane (tricyclo[1.1.1.0¹,³]pentane) **1**), final reaction concentration = 0.2 M with a 1:1 ratio of Et₂O:DME. The vial was then cooled in an ice bath (0 °C) and the solution was degassed by sparging with argon for 15 min. The vent needle was removed and [1.1.1]propellane **1** (1.0 equiv.) and aldehyde (1.5 equiv., if liquid) were then added. The vial was then capped and double-sealed with parafilm. The reaction mixture was then placed in a cooling bath at 10 °C and irradiated with blue LEDs for 24 h. The reaction mixture was then concentrated in vacuo. MeOH (5 mL/mmol of **1**) and CH₂Cl₂ (5 mL/mmol of **1**) were added to the vial and the reaction mixture cooled to –78 °C. NaBH₄ (5.0 equiv.) was added and the reaction mixture was stirred for 30 min to 1 h at –78 °C. The reaction was quenched by addition of NH₄Cl (aq., sat., 5 mL/mmol of **1**) and water (5 mL/mmol of **1**). The phases were separated, and the aqueous phase was extracted with CH₂Cl₂ (×3). The combined organic extracts were dried (Na₂SO₄) and concentrated in vacuo. The residue was purified by flash column chromatography (pentane/ether eluent). The enantiomeric excesses (% ee) were determined by derivatization of the alcohol product as a 4-dimethylaminobenzoate ester, which was analyzed by high-performance liquid chromatography using chiral stationary phases (CHIRALPAK IB or IC column).

## Data availability
The authors declare that the data supporting the findings of this study are available within the paper and its Supplementary Information files. All other requests for materials and information should be addressed to the corresponding authors.

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

## Acknowledgements
M.L.J.W. and A.J.S. thank the EPSRC Centre for Doctoral Training in Synthesis for Biology and Medicine for studentships (EP/L015838/1), generously supported by AstraZeneca, Diamond Light Source, Defence Science and Technology Laboratory, Evotec, GlaxoSmithKline, Janssen, Novartis, Pfizer, Syngenta, Takeda, UCB, and Vertex.

E.A.A. thanks the EPSRC for support (EP/S013172/1). A.J.S and F.D. thank the EPSRC Centre for Doctoral Training for Theory and Modelling in Chemical Sciences (EP/L015722/1) for providing access to the Dirac cluster at Oxford. This work used the Cirrus UK National Tier-2 HPC Service at EPCC (http://www.cirrus.ac.uk) funded by the University of Edinburgh and EPSRC (EP/P020267/1). We thank K. G. Leslie for assistance with the photophysical studies.

## Author contributions

M.L.J.W. and E.A.A. conceived the work. M.L.J.W. designed and carried out the synthetic experiments. A.J.S. and F.D. designed and carried out the theoretical work. A.J.S. and M.L.J.W. designed and carried out the photophysical experiments. M.L.J.W., A.J.S., J.J.M., F.D., and E.A.A. analyzed the data, and discussed and co-wrote the manuscript.

## Competing interests

The authors declare no competing interests.
