## [Peer Review File · Nature Communications]

REVIEWER COMMENTS

Reviewer #1 (Remarks to the Author):

All issues have been addressed satisfactorily. Accept.

Reviewer #2 (Remarks to the Author):

I already had the opportunity to review this work in a previous submission round and I didn't change my positive feedback related to the present contribution. Accordingly, I am highly supportive of publication of this ms in Nat. Commun.

While the Authors have addressed satisfactorily most of the issues/comments raised by the reviewers, I think that they failed to address the request related to redox potentials for several reasons. In my opinion, the reported data and the corresponding assumptions are partially wrong and careful revision is therefore recommended.

- As for the oxidation of enamine by the excited state photocatalyst, the Authors have probably used a wrong sign because the ΔE should be: $0.66 - 0.77 = -0.11$. This step seems to be slightly endergonic. The indicated redox potential for the substrate should be: $E[\alpha\text{-iminyI radical cation/propanal enamine}] = +0.77$ V vs. SCE. In addition, please avoid using the "E_cell" tag, but simply use: ΔE .

- I don't understand why the Authors wrote " $E^{\circ}_{\text{ox}}[\text{Ir(II)/Ir(III)}] = +1.51$ V" instead of " $E[\text{Ir(III)/Ir(II)}] = -1.51$ V", in a way similar to what reported in Ref. 46.

- I don't see the reason to cite Ref. 47 instead of the actual article wherein the redox potential has been determined, see: J. Phys. Chem. 1993, 97, 1610.

Reviewer #1: (Remarks to the Author):

All issues have been addressed satisfactorily. Accept.

We thank the Reviewer for this supportive view!

Reviewer #2 (Remarks to the Author):

I already had the opportunity to review this work in a previous submission round and I didn't change my positive feedback related to the present contribution. Accordingly, I am highly supportive of publication of this ms in Nat. Commun.

While the Authors have addressed satisfactorily most of the issues/comments raised by the reviewers, I think that they failed to address the request related to redox potentials for several reasons. In my opinion, the reported data and the corresponding assumptions are partially wrong and careful revision is therefore recommended.

- As for the oxidation of enamine by the excited state photocatalyst, the Authors have probably used a wrong sign because the ΔE should be: $0.66 - 0.77 = -0.11$. This step seems to be slightly endergonic. The indicated redox potential for the substrate should be: $E[\alpha\text{-iminyl radical cation/propanal enamine}] = +0.77$ V vs. SCE. In addition, please avoid using the "E_cell" tag, but simply use: ΔE .

Our response: We thank the Reviewer for noticing the mistake in the reduction potential of the enamine/ α -enaminy radical cation couple, and apologise for reproducing the error in sign that was also made in Reference 48. The text has been modified to reflect this change, and the nomenclature has been updated throughout, as suggested. The two text modifications are as follows:

As discussed above, we propose that the mechanism of this reaction involves initial oxidation of the enamine to the α -iminyl radical cation by the excited state

Ir(III) complex ($E^\circ[\text{Ir(III)}^*/\text{Ir(II)}] = +0.66$ V vs. SCE;⁴⁶ $E^\circ[\alpha\text{-iminyl radical cation/propanal enamine}] = +0.77$ V vs. SCE;⁴⁸ $\Delta E^\circ = -0.11$ V).

- I don't understand why the Authors wrote " $E^\circ_{\text{ox}}[\text{Ir(II)}/\text{Ir(III)}] = +1.51$ V" instead of " $E[\text{Ir(III)}/\text{Ir(II)}] = -1.51$ V", in a way similar to what reported in Ref. 46.

Our response: We are happy to alter this formatting and have modified the text as follows, in three places:

Addition of the α -iminyl radical cation to **1** leads to a bicyclopentyl radical, which abstracts a hydrogen atom from the thiol HAT catalyst to deliver the BCP product; the thiyl radical that results is a competent species for catalyst turnover ($E^\circ[\text{PhS}^\bullet/\text{PhS}^-] = +0.16 \text{ V vs. SCE}$; ⁴⁹ $E^\circ[\text{Ir(III)}/\text{Ir(II)}] = -1.51 \text{ V vs. SCE}$; ⁴⁶ $\Delta E^\circ = +1.67 \text{ V}$).

...this reaction presumably proceeds by attack of the BCP radical on the S–N bond, leading to a phthalimidyl radical that is also capable of catalyst turnover (E°

$[\text{Ir(III)}/\text{Ir(II)}] = -1.51 \text{ V vs. SCE}$; ⁴⁶ $E^\circ[\text{SuccN}^\bullet/\text{SuccN}^-]$ of the related succinimidyl radical has been measured as $+1.98 \text{ V vs. SCE}$).⁴⁷

- I don't see the reason to cite Ref. 47 instead of the actual article wherein the redox potential has been determined, see: J. Phys. Chem. 1993, 97, 1610.

Our response: Ref. 47 has been replaced as suggested, and the value of $E^\circ[\text{SuccN}^\bullet/\text{SuccN}^-]$ has been corrected from $+1.96 \text{ V}$ to $+1.98 \text{ V}$ in the text, according to the updated reference.